# uvCLAP is a fast and non-radioactive method to identify in vivo targets of RNA-binding proteins

Daniel Maticzka[1], Ibrahim Avsar Ilik[2], Tugce Aktas[2], Rolf Backofen (ID) [1,3] & Asifa Akhtar (ID) [2]

RNA-binding proteins (RBPs) play important and essential roles in eukaryotic gene expression regulating splicing, localization, translation, and stability of mRNAs. We describe ultraviolet crosslinking and affinity purification (uvCLAP), an easy-to-use, robust, reproducible, and high-throughput method to determine in vivo targets of RBPs. uvCLAP is fast and does not rely on radioactive labeling of RNA. We investigate binding of 15 RBPs from fly, mouse, and human cells to test the method's performance and applicability. Multiplexing of signal and control libraries enables straightforward comparison of samples. Experiments for most proteins achieve high enrichment of signal over background. A point mutation and a natural splice isoform that change the RBP subcellular localization dramatically alter target selection without changing the targeted RNA motif, showing that compartmentalization of RBPs can be used as an elegant means to generate RNA target specificity.

[1] Bioinformatics Group, Department of Computer Science, University of Freiburg, Georges-Koehler-Allee 106, 79110 Freiburg, Germany. [2] Max Planck Institute of Immunobiology and Epigenetics, Stuebeweg 51, 79108 Freiburg, Germany. [3] Centre for Biological Signalling Studies (BIOSS), University of Freiburg, Schaenzlestr. 18, 79104 Freiburg, Germany. These authors contributed equally: Daniel Maticzka, Ibrahim Avsar Ilik, Tugce Aktas. Correspondence and requests for materials should be addressed to R.B. (email: backofen@informatik.uni-freiburg.de) or to A.A. (email: akhtar@ie-freiburg.mpg.de)

Transcriptional control of gene expression is a highly regulated and intensely studied phenomenon that requires a plethora of DNA-interacting proteins and other upstream factors that integrate intracellular and extracellular information to effect an appropriate transcriptional response[1,2]. The message of an RNA can be changed, muted, enhanced, localized, or delayed post-transcriptionally through the collective action of many RNA-binding[3] and RNA-modifying proteins[4] that act on these RNA transcripts via binding to specific RNA elements[5].

CLIP-Seq approaches, such as HITS-CLIP[6], iCLIP[7], PAR-CLIP[8], and CRAC[9], have become the prevalent experimental method for determining binding sites of RNA-binding proteins (RBPs) in living cells. To date, CLIP-Seq has been used to determine binding sites of more than 100 distinct RBPs[10,11]. These experiments lead to the recent elucidation of the complex nature of RBP-mediated post-transcriptional regulation. To further uncover the intricate regulatory relationships in the cell, where multiple RBPs can compete or cooperate with each other for target site selection[12], researchers will need to probe multiple RBP isoforms, to investigate RBPs with modified or disabled functionalities, to examine multiple candidates involved in a regulatory process, and also to compare results from multiple cell types and knockdown/overexpression scenarios. Widespread application of current CLIP-Seq methods in this manner is generally thwarted by the common use of radioactive labeling of crosslinked RNA, lengthy protocols, and dependence on the availability of high-quality antibodies.

Here, we present ultraviolet crosslinking and affinity purification (uvCLAP), a method for identifying in vivo binding sites of RBPs that is fast, robust, does not rely on radioactivity, and that allows to quantify the amount of nonspecific background to obtain transcriptome-wide high-resolution RNA–protein interaction maps with high specificity. We evaluate uvCLAP on the three major model organisms *Homo sapiens*, *Mus musculus*, and *Drosophila melanogaster* and perform experiments for a wide variety of RBPs, namely the KH domain-containing RBPs QKI-5, QKI-6, KHDRBS1-3, and hnRNPK, the DEAD-box helicases eIF4A1 and EIF4A3, the DExH-Box helicases MLE[13] and DHX9[14], the member of the exon junction complex (EJC) MAGOH, and the mouse MSL complex members MSL1 and MSL2. In addition, we probe mutant constructs of KHDRBS1, KHDRBS2, and MLE. In total, we analyze 23 uvCLAP experiments (investigating 15 RBPs) in human, mouse, and fly.

## Results

**uvCLAP as an effective method to determine RBP targets**. The hallmark of most CLIP-seq approaches is radioactive labeling of RNA that is covalently bound to a protein. Since proteins tend to run at discrete positions on a sodium dodecyl sulfate (SDS)/lithium dodecyl sulfate-polyacrylamide gel electrophoresis setup, cutting out the labeled region removes contaminating RNA and RNA–protein complexes as they migrate to different regions of the gel. In addition, the crosslinked protein–RNA complex is often electrophoretically transferred to a nitrocellulose membrane, which typically binds to proteins (and consequently protein–RNA adducts) but not to free RNA. This transfer is meant as an additional step for removing non-crosslinked RNA but can also introduce additional contaminating RNA[15].

We reasoned that a stringent tandem affinity purification protocol would circumvent these time-consuming and critical steps while nonetheless removing contaminating free RNA and proteins that might nonspecifically co-purify with the tagged proteins. To this end, we decided to use the His$_6$-biotinylation sequence-His$_6$ tandem (HBH) tag[16] that allows rapid and ultra-

clean purifications without the use of antibodies. We also added a 3xFLAG tag right before the HBH tag to increase the versatility of the construct, which we will refer to as the 3FHBH tag. In contrast to antibodies, tagged constructs can be expected to have similar pulldown efficiency, leading to improved comparability across multiple conditions. To reveal the genomic origin of nonspecific background, we decided to include background controls employing mock pulldowns, using the expression vector carrying the 3FHBH tag without a gene insert. As a straightforward method for capturing the quantitative relationship between pulled down RNA and nonspecific background, we combined signal and control libraries prior to amplification and sequencing.

The complete uvCLAP protocol from cells-on-plates to sequencing libraries takes 4 days with the use of a single gel purification step (Fig. 1a and Supplementary Fig. 1a, b). Briefly, lysates from cell lines expressing RBPs of interest and control cell lines are prepared and subjected to a fast tandem affinity purification. First, paramagnetic beads that bind to polyhistidine tagged proteins in <10 min are used to partially purify the protein of interest while removing endogenously expressed biotinylated proteins (lane 2 in Fig. 1b). After eluting the bound protein with imidazole, a second, more stringent streptavidin purification is carried out (lane 3 in Fig. 1b). This is then followed by partial RNase digestion using RNaseI, repairing the ends of digested RNA with T4 polynucleotide kinase (PNK), ligation of adapters, reverse transcription with barcoded reverse transcription primers, mixing of all samples and separation of complementary RNA (cDNA) products on a denaturing polyacrylamide (PAA) gel, circularization of cDNA, linearization of circular products, and finally PCR amplification (Fig. 1a). uvCLAP profiles of various RBPs expressed in human cells show characteristic binding on their target RNAs. We also observed very weak and dispersed events from the control libraries, indicating a successful removal of free or nonspecifically bound RNA by the tandem affinity pulldown (Fig. 1c, d and Supplementary Fig. 1b).

Early mixing of cDNA generated from mock pulldowns and cDNA generated from proteins of interest allowed us to thoroughly evaluate the effects of the tandem purification. Multiplexing was implemented using a triple-tag strategy that allowed us to distinguish pulldown conditions, biological replicates, and size fractions. To improve the precision of our measurements, we used unique molecular identifiers (UMIs) for detecting individual crosslinked RNAs. Since the quantitative relationship between signal and control becomes unclear when samples are independently amplified and sequenced, the multiplexed libraries were subjected to joint amplification and sequencing. This also allowed us to circumvent an additional challenge typically arising with the empirical measurement of CLIP-Seq background controls: the combination of ineffectual pulldown and stringent washing typically results in small library sizes requiring high numbers of PCR cycles in order to be suitable for a dedicated sequencing run[17]. Taken together, this setup should preserve the quantitative relationship between multiplexed uvCLAP samples (Fig. 2a).

**uvCLAP preserves RNA quantities of multiplexed samples**. The quantitative relationship between samples is preserved, if the proportions of observed events match the proportions of RNA in the underlying samples. To determine to what extent the multiplexed treatment of specific and nonspecific pulldown conditions would preserve the quantity of detected RNA, we compared the total number of crosslinking events between pairs of biological replicates, each sharing the respective pulldown condition and construct expression. In total, we compared 23 specific and 7 nonspecific conditions in 6 multiplexed uvCLAP runs in *Homo*

*sapiens*, *Drosophila melanogaster*, and *Mus musculus* (Fig. 2b and Supplementary Table 1). This revealed a highly significant agreement of the number of events between paired replicates (Supplementary Fig. 2b, Pearson's correlation 0.997, $n = 30$, two-sided, $p$ value < 2.2e−16), indicating that uvCLAP events capture the amount of pulled down RNA exceptionally well.

We next asked, to what extent the quantitative relationship is preserved at the binding site level. Therefore, we compared peaks, called by slicing the genome into bins, between each pair of biological replicates. For all 100 nucleotide bins covered by at least two events in both replicates, we calculated fold changes between the replicates (Fig. 2c and Supplementary Fig. 3). The

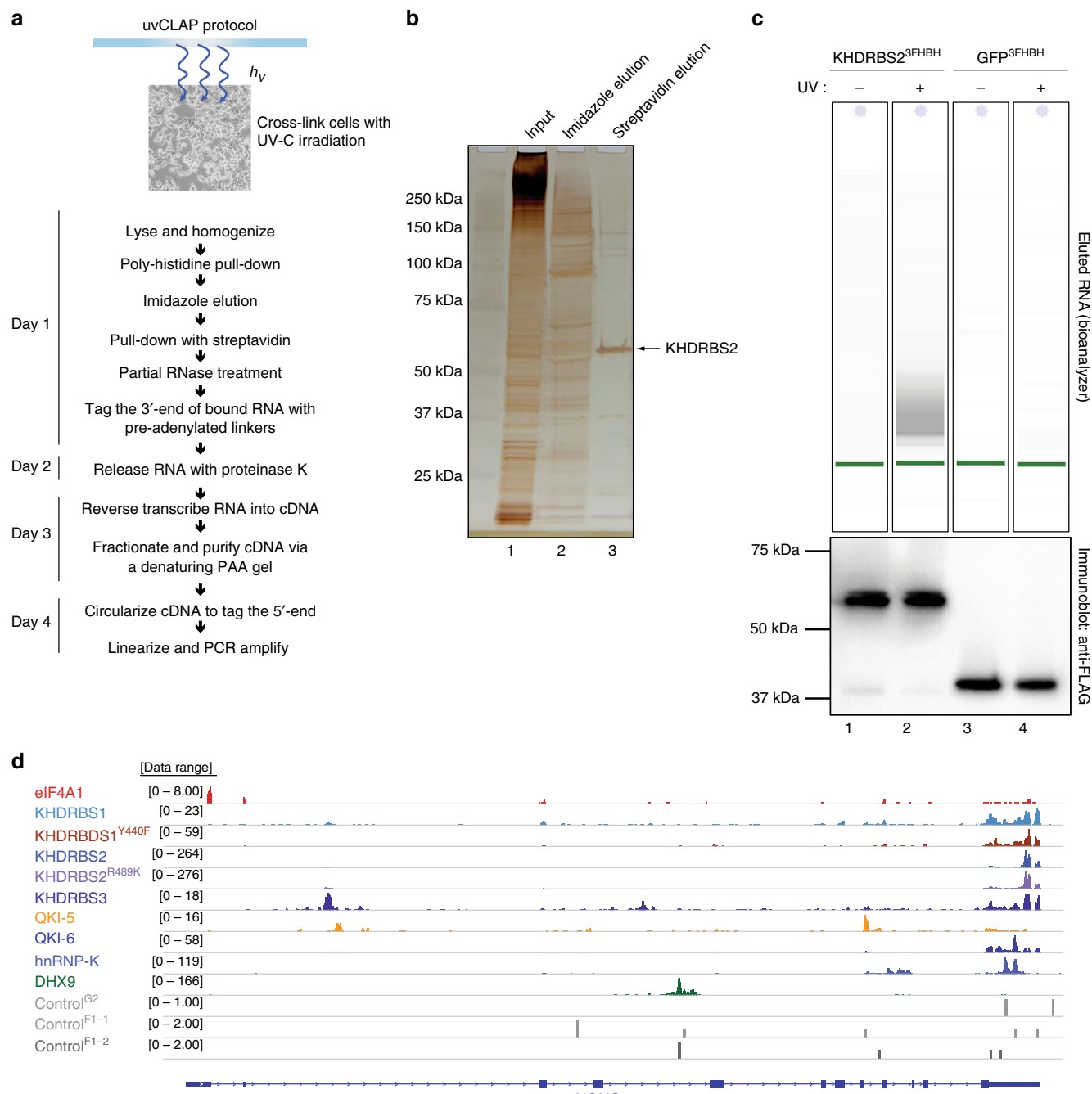

**Fig. 1** uvCLAP identifies in vivo targets of RBPs. **a** Experimental workflow of uvCLAP, starting from cells-on-plates to the generation of sequencing libraries. **b** Silver staining analysis of tandem affinity purified of 3xFLAG-HBH tagged KHDRBS2 under the highly stringent uvCLAP conditions (see Methods). Lane 1: initial lysate, lane 2: eluate after the first step of purification, lane 3: final eluate from streptavidin beads. **c** Bioanalyzer traces of RNA purified from non-crosslinked KHDRBS2^3FHBH (lane 1), UV-crosslinked KHDRBS2^3FHBH (lane 2), non-crosslinked GFP^3FHBH (lane 3) and UV-crosslinked GFP^3FHBH (lane 4) lysates at the end uvCLAP purification as in **a** and **b** (top). Eluted protein from the same samples, analyzed in parallel via immunoblotting with anti-FLAG antibodies (bottom). Also see Supplementary Fig. 2b for a comparison to FLAG-only purifications. **d** IGV (Integrative Genomics Viewer) snapshot of uvCLAP profiles for eIF4A1, KHDRBS1, KHDRBS1^Y440F, KHDRBS2, KHDRBS2^R489K, KHDRBS3, QKI-5, QKI-6, hnRNPK and DHX9 on NONO gene. Biological replicates are merged for this representation, data range represents the coverage of uvCLAP reads. Only plus strand data is represented for visual clarity

fold changes of all pairs of replicates were consistently centered near zero, indicating a robust between-replicate agreement. The amount of noise was low for bins with high crosslinking counts and steadily increased for bins with lower counts, a characteristic commonly observed with RNA-Seq data[18].

We then wondered what library normalization—if any—would be necessary for uvCLAP. Library normalization is commonly applied to compensate for differences in library preparation when comparing different RNA-Seq[19] or CLIP-Seq[20] samples. Having matched amplification and sequencing conditions of multiplexed samples, we should be able to dispose with the need for library

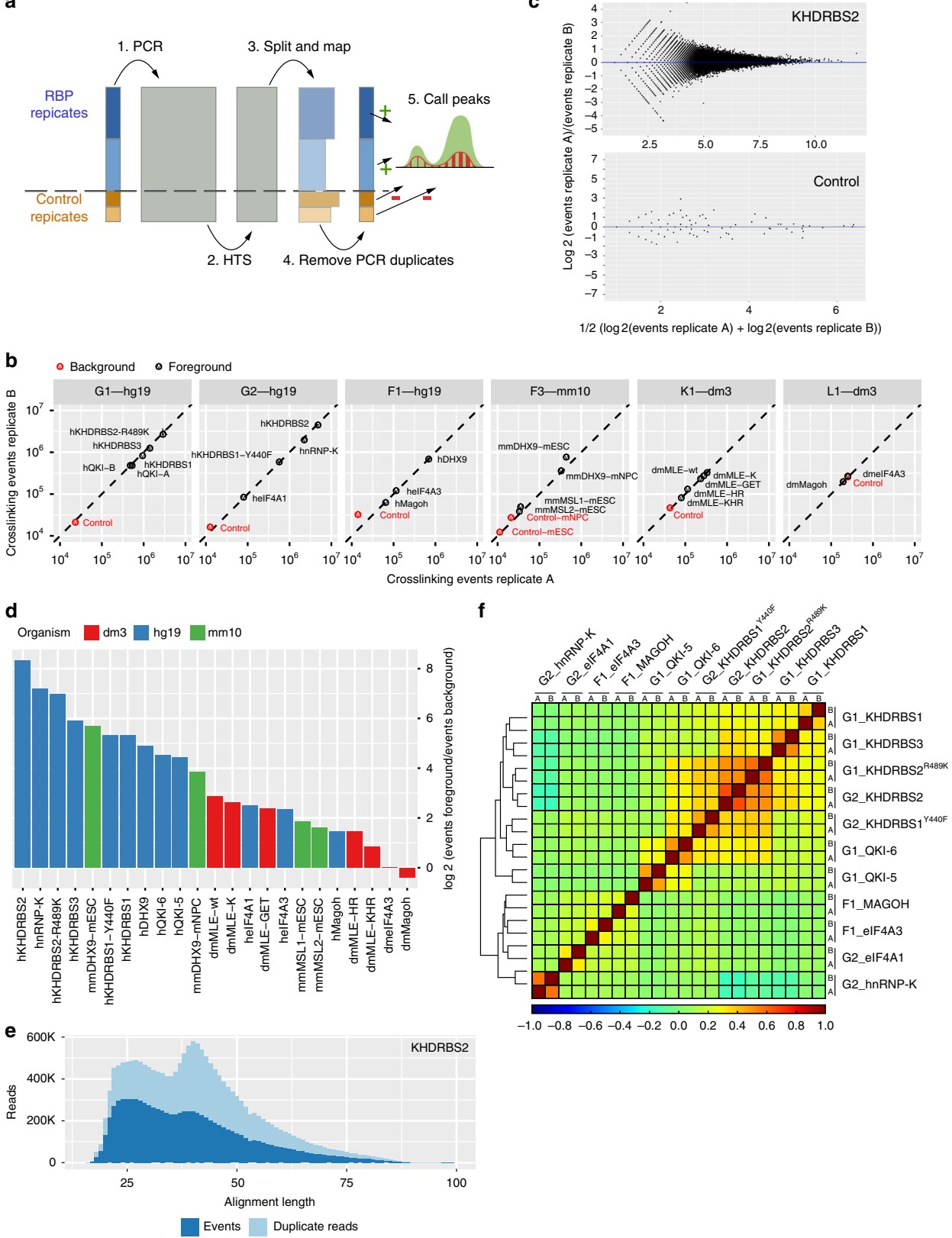

normalization. To investigate to what extent library normalization would be required, we determined library normalization factors for uvCLAP biological replicates using the median ratios of counts method[18,21] (Supplementary Table 2). Normalization factors for 23 of the 30 pairwise replicates were 1, indicating that no library normalization is required. The average normalization factor across all replicates was 0.96, showing that uvCLAP counts accurately reflect RNA quantities on the site level.

We also investigated to what extent the barcodes used for distinguishing pulldown conditions would influence quantification. A comparison of the six sets of signal and control events from library K1 that were located on the 18S ribosomal RNA pseudogene CR41602, which is a common source of nonspecific RNA for CLIP-Seq experiments in *Drosophila*, revealed a relative standard deviation of only 12.10%[13], indicating that uvCLAP barcodes are unlikely to introduce a bias during sequencing or amplification.

In summary, uvCLAP joint amplification and sequencing preserves the quantitative relationship between samples both on the level of total library RNA and local site counts.

**Enrichment over control is specific to the targeted RBP.** Having established that uvCLAP events accurately account for the amount of RNA in specific and nonspecific libraries, we investigated the relationship between signal and control events in our libraries. To allow the direct comparison of pulldown conditions on the level of detected events, the comparison of RNA amounts in the underlying libraries must be meaningful. This is not an issue when comparing signal and control libraries because the expression of empty constructs is not directly linked to the amount of pulled down RNA, but it imposes additional constraints for the comparison of specific pulldown conditions (most notably matched RBP expression and pulldown efficiency).

To get an overview of the global relationship between events in signal and control libraries, we calculated the total enrichment of specific versus nonspecific conditions based on the total number of crosslinking events per library (replicates combined) (Fig. 2d and Supplementary Table 3). This analysis revealed an over three orders of magnitude variance in the library-wide enrichment of signal over control libraries. At the high end, we observed a more than 300-fold enrichment over controls for human KHDRBS2 (a large library with 9.3 million crosslinking events). At the low end, we found only a very slight enrichment over controls for *Drosophila* EIF4A3 (signal-to-control ratio 1.02) and no enrichment for *Drosophila* MAGOH (signal-to-control ratio 0.77), which are two small libraries with 517,000 and 392,000 crosslinking events. Experiments for most proteins achieved

more than 14-fold enrichment over the controls (hsDHX9, mmDHX9, QKI-5, QKI-6, KHDRBS1-3, and HNRNPK).

In case of the fly experiments, we observed an increase in the absolute number of control events (Fig. 2b), which was most salient for the control of library L1 containing dmMAGOH and dmEIF4A3. These two samples showed no appreciable enrichment of events compared to the control, suggesting that binding of dmMAGOH and dmEIF4A3 must be evaluated carefully. A further evaluation of expected binding at the upstream of exon–exon junctions, which is discussed in detail below, confirmed our assessment of these quality indicators. Despite the similar amounts of events in signal and control libraries, an appreciable number of events could be extracted. This was most likely caused by deeper sequencing as a result of the small sample sizes and shows that a high absolute number of crosslinking events is an insufficient indicator of a highly specific CLIP-Seq experiment.

We also detected an above-average absolute number of control events for *Drosophila* library K1, which combined the RNA helicase MLE and several of its mutants. MLE has a unique mode of binding and predominantly targets only two noncoding RNAs[22], one of which was expressed in the cell line used for the uvCLAP experiments. Binding to a single gene necessarily leads to small libraries, elevating the absolute counts of control events. Nonetheless, uvCLAP performed well in this unique and challenging setting, which allowed us to thoroughly investigate the functional unit necessary for dosage compensation in *Drosophila*[13].

Our observation that uvCLAP nonspecific background appears more pronounced for RBPs with small library sizes, matches, and extends the results from the evaluation of nonspecific background for PAR-CLIP by Friedersdorf and Keene[23]. While low absolute amounts of unspecific background are a necessary condition for achieving high specificity with CLIP-Seq, the pulled down RNA must also be sufficiently enriched over the background to obtain meaningful results. Our results show that low background levels relative to the signal can only be presumed under ideal conditions, that is, for proteins that crosslink well and bind to a large number of sites.

We conclude that the quantification of nonspecific background is required to reliably determine the binding of putative RBPs using CLIP-Seq. Additionally, global enrichment over background controls can serve as a straightforward means to identify problematic experiments; the extent of background relative to signal must be checked on a case-by-case basis.

**cDNA length-specific amplification bias is mitigated by UMIs.** Having separately tagged size fractions available, we observed that

---

**Fig. 2** uvCLAP is a quantitative and reproducible assay. **a** Combined amplification and sequencing of multiplexed uvCLAP libraries preserves relative quantities of signal and control libraries. Areas of light blue boxes indicate amounts of RNA, cDNA, and numbers of alignments and events. 1. After reverse transcription, libraries are combined and subjected to PCR amplification. 2. High-throughput sequencing determines nucleotide-sequences of a subset of cDNAs. 3. Reads are assigned to respective libraries according to barcodes and mapped to the genome. 4. Reads are merged into crosslinking events according to unique molecular identifiers (UMIs), mitigating bias from PCR amplification. 5. Peak calling utilizes information from the controls to disregard regions not enriched over background (depicted by the minus symbol in red and plus symbol in green). **b** Comparison of the total number of crosslinking events identified for pairwise biological replicates of 23 pulldown conditions (black) and 7 nonspecific controls (red). **c** MA-plots comparing crosslinking event counts for pairwise biological replicates of KHDRBS2 and the corresponding background control for genomic 100 nucleotide bins covered by at least 2 crosslinking events in both replicates. The median log2 fold change is indicated in blue (see Supplementary Fig. 3 for the full set of plots for all pulldown conditions). **d** Log2-ratios of crosslinking events to nonspecific events from background controls for 23 pulldown conditions. **e** Number of reads categorized as crosslinking events and PCR duplicates dependent on alignment length as proxy for cDNA insert size (see Supplementary Fig. 4 for the full set of plots for all pulldown conditions). **f** Clustered heatmap of pairwise Spearman correlations (deeptools2) for crosslinking events of uvCLAP replicates on 764,727 merged JAMM peak regions for human KHDRBS1-3, KHDRBS1[Y440F], KHDRBS1[R489K], QKI-5, QKI-6, MAGOH, eIF4A1, EIF4A3 and hnRNPK. Clusters were joined using the Nearest Point Algorithm

events from high-size fractions (H, cut around 75 nt) and mid-size fractions (M, cut around 50 nt) were represented by more duplicate reads than the short-size fractions (L, cut around 25 nt), which may indicate a length bias introduced by PCR amplification[24]. This effect was very pronounced for the high-size fractions to the extent that only few crosslinking events could be identified despite a large number of reads. In consequence, we decided not to use the high-size fractions and omitted their sequencing for *Drosophila* EIF4A3 and MAGOH uvCLAP.

To further investigate the influence of cDNA length on the amount of duplicate sequences, we evaluated the number of uniquely aligned reads and resulting events of the low-size and mid-size fractions in relation to the length of the corresponding alignments (Fig. 2e and Supplementary Fig. 4). The resulting distributions frequently revealed an excess of PCR duplicates for cDNAs in the range of 40–60 nts, which appear to have been amplified more efficiently than shorter or longer cDNAs. This length bias largely disappeared at the level of crosslinking events, leading to mostly unbiased distributions over the full range of cDNA sizes. In consequence, uvCLAP maintains a wide range of cDNA lengths, which allows to efficiently delineate the complete RNA-binding sites[25].

**Biological replicates in uvCLAP show high correlation**. To obtain a high-level overview on similarities and differences between different pulldown conditions, biological replicates, and controls, we merged the JAMM[26] peaks (Supplementary Table 4) of human proteins (KHDRBS1-3, KHDRBS1$^{Y440F}$, KHDRBS2$^{R489K}$, QKI-5, QKI-6, eIF4A1, EIF4A3, MAGOH, and hnRNPK) and calculated Spearman's correlations based on the counts of crosslinking events located on the resulting 764,727 non-overlapping regions (Fig. 2f). All correlations between biological replicates were higher than correlations between unrelated pulldown conditions. The average Spearman's correlation between biological replicates of 0.46 ($N = 764,728$; two-sided; all $P$ values < 0.00001) was much higher than the average correlation of 0.08 between unrelated replicates (excluding comparisons between KHDRBS1-3 and between QKI-5 and QKI-6). We also observed high correlations between replicates of KHDRBS2 and its pseudo-mutant construct KHDRBS2$^{R489K}$, but not between replicates of KHDRBS1 and its mutant KHDRBS1$^{Y440F}$ that differ in cellular localization (see Fig. 3a–h for cellular localization of the mutants and isoforms and below for a detailed discussion of the mutant constructs).

The above analysis provided a general overview at the expense of including many regions not relevant for the evaluation of a given uvCLAP experiment. For this reason, we also evaluated a practice-oriented setting and compared crosslinking events for each pair of biological replicates. To account for the increased variability of crosslinking events in regions with lower counts of crosslinking events (Fig. 2c and Supplementary Fig. 3), we used the smaller sets of high-confidence peaks calculated by PEAKachu[27] (Supplementary Table 5). In this setting, the average Spearman's correlation between biological replicates increased to 0.92 (Supplementary Table 5). The average Spearman's correlation between unrelated replicates was much lower at 0.19 (each pair of unrelated replicates was evaluated using half of the merged regions of the corresponding pair of peaks).

**uvCLAP background is independent from signal**. To obtain a high-level view on uvCLAP background, we determined the distributions of control crosslinking events on different classes of genes (Supplementary Fig. 2c–e). Here, the two fly control libraries exhibited similar distributions over the targeted classes of genes, the most prominent class of targets being ribosomal RNAs.

The two mouse libraries also showed similar distributions, many events were located on intronic and intergenic regions. The three human controls (G1, G2, and F1) had similar low number of events located on ribosomal RNA, noncoding RNA, and pseudogenes (Supplementary Fig. 2c). We observed a larger variability of the distributions on regions of coding genes, intergenic and antisense regions in comparison to fly and mouse controls (Supplementary Fig. 2c–e). This variability may be due to the stochastic nature of nonspecific background events; however, we cannot exclude the possibility of a weak shadowing of one of the signal samples in the control libraries.

We next evaluated the connection between uvCLAP signal and nonspecific background by determining the number of JAMM peaks more than 50 nucleotides apart from any read of the corresponding control (Supplementary Table 4). The 94.15% of peaks derived for the 23 pulldown conditions were located far from the control events obtained; 16 of 23 pulldown conditions had more than 80% of peaks without evidence of surrounding control reads. Proteins with higher correspondence to control reads either are known to inefficiently crosslink (hMAGOH), had low numbers of binding sites (mmMSL1 and mmMSL2), or were mutant constructs with one or two disabled double-stranded RNA-binding domains (dmMLE-HR and dmMLE-KHR). The largest overlap with the controls was observed for dmEIF4A3 and dmMAGOH with 26.21% and 16.39% of peaks in the vicinity of control reads. The overwhelming majority of sites detected by uvCLAP were independent of nonspecific background. The seven proteins with the highest overlap between peaks and reads from the controls also had the lowest global signal-to-control ratios (below 5).

Since control events generally exhibited low overlap with uvCLAP peak regions, we decided to independently evaluate the expression of our six replicates of human background controls and calculated Spearman's correlations for the number of control events on 106,770 bins of 100 nt length that overlapped at least one background event, considering all possible pairs of control replicates. This evaluation resulted in an average Spearman's correlation of −0.16, indicating that human uvCLAP background is mostly of a stochastic nature.

For both signal and control libraries of the *Drosophila* experiments, we found events on the ribosomal RNA pseudogene CR41602[13]. This observation appears to be common for fly experiments since events on this RNA were observed for CLIP-Seq libraries from different cell types[13,22], different CLIP-Seq methods[13,22], and different labs[22,28].

**uvCLAP recovers binding preferences of EIF4A1 and EIF4A3**. We evaluated uvCLAP using the two highly related DEAD-box helicases EIF4A1 and EIF4A3. We chose these proteins for two reasons: first, both proteins have well-defined RNA-binding behavior. EIF4A1 is part of the cytoplasmic EIF4 complex that scans the 5′-UTRs of mRNA in search of a start codon, thus a strong 5′-UTR enrichment would indicate a successful experiment. EIF4A3, on the other hand, is part of the predominantly nuclear EJC that binds 20–30 nt upstream of exon–exon junctions, therefore we expected to detect an exonic enrichment and a positional bias relative to exon junctions. Second, EIF4A3 interacts mainly with the sugar-phosphate backbone of the target RNA[29,30], which makes it a poor UV crosslinking protein and hence a challenging protein to use as a benchmark[30].

As expected, because of its membership in the eIF4 complex, 5′-UTRs were the most abundant type of eIF4A1 targets (Fig. 3a). In total, 63% of the 2653 eIF4A1 JAMM peaks located on protein-coding gene regions were annotated as 5′-UTRs. For human EIF4A3 and MAGOH, coding exons were the most abundant

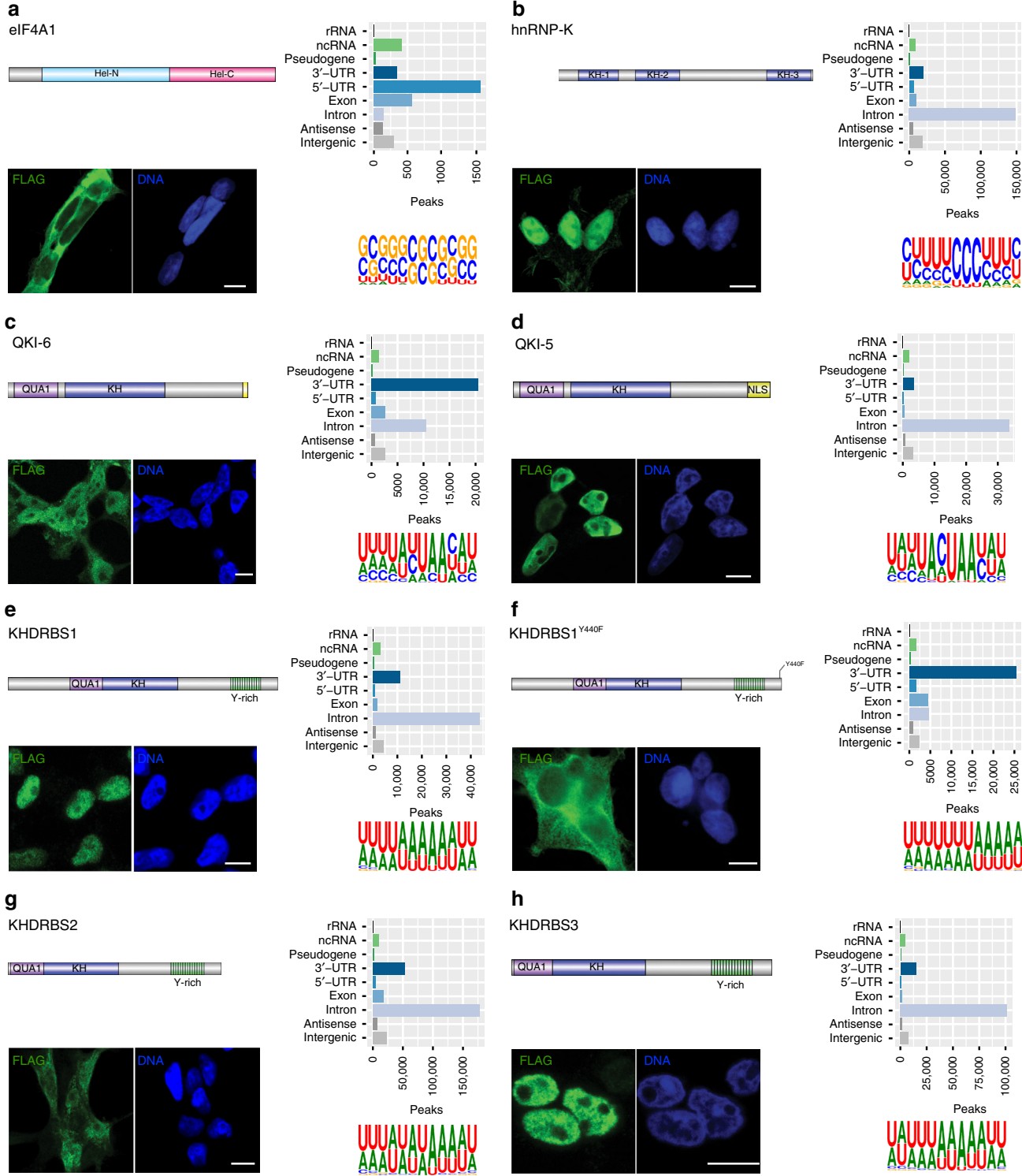

**Fig. 3** Cellular localization of RBPs do not change their preferred RNA motif. Cellular localization of FLAG-tagged RBPs (scale bars represent 10 μm); number of JAMM peaks located on genomic target classes with each peak assigned once to the category with highest priority corresponding to the order rRNA, ncRNA, pseudogene, 3′-UTR, 5′-UTR, exon, intron, antisense, intergenic (see Supplementary Table 6 for the counts of peaks on genomic targets); and Sequence motifs determined by GraphProt[36] for human proteins **a** EIF4A1, **b** hnRNPK, **c** QKI-6, **d** QKI-5, **e** KHDRBS1, **f** KHDRBS1[Y440F], **g** KHDRBS2, and **h** KHDRBS3

type of targets (Fig. 4a), with 78% of hsEIF4A3 and 81% of hsMAGOH JAMM peaks located on protein-coding genes targeting regions annotated as coding exons.

To accurately determine the binding of human and fly EJC (EIF4A3 and MAGOH) near exon–exon junctions, we realigned the libraries using the splice-aware mapper HISAT2[31,32], performed the same processing as done for the other uvCLAP sets and ascertained the locations of crosslinked nucleotides relative to 5′-exon and 3′-exon ends. We observed an enrichment of crosslinked nucleotides for human EIF4A3 upstream of

exon–exon junctions (Fig. 4b), a pattern consistent with its role as a member of the EJC. In contrast, crosslinked nucleotide positions from the controls were uniformly distributed in relation to exon–exon junctions. For MAGOH we observed a similar pattern that was shifted slightly upstream to that of EIF4A3.

We obtained very similar results for *Drosophila* EIF4A3. Sixty-eight percent of the peaks located on protein-coding genes were annotated as coding exons (Fig. 4c). We also observed a positional pattern similar to hsEIF4A3 (Fig. 4d). In contrast to dmEIF4A3, *Drosophila* MAGOH peaks were only slightly enriched for coding exons. The binding profile relative to exon–exon junctions was uniform in the regions 200 nucleotides upstream and downstream of exon–exon junctions, similar to the profile of the background control.

In summary, uvCLAP recovered the known position-specific binding of DEAD-box helicase eIF4A1 on 5′-UTRs and of the EJC members hsEIF4A3 and hsMAGOH near exon junctions. The binding pattern of dmMAGOH was independent of exon–exon junctions. This indicates that no binding was detected for dmMAGOH and further validates our initial assessment of the low signal-to-control ratio and the increased occurrence of control events on signal peaks. This result highlights the

usefulness of quantitative background controls to assess the success of an experiment at an early stage of the analysis, which will be especially helpful when targeting RBPs with uncharted binding preferences.

**uvCLAP recovers binding preferences of KH domain proteins.** Next, we evaluated QKI, a protein with a previously identified sequence specificity, that is known to regulate target-mRNA stability[33] and circRNA formation[34]. QKI has multiple isoforms that slightly differ at their C termini, but nonetheless have a profound effect on its subcellular localization; the shorter isoforms lack a nuclear localization signal (NLS) that makes them predominantly cytoplasmic[35].

We chose to evaluate the predominantly nuclear isoform QKI-5 as well as the shorter isoform QKI-6, which is reported to be both nuclear and cytoplasmic[34]. We could confirm this behavior for our triple-tagged constructs via anti-FLAG staining (Fig. 3c, d). In accord with this observation, peaks of the nuclear isoform QKI-5 were located predominantly on intronic regions, whereas peaks of QKI-6 were located to a large extent on 3′-UTRs as well as intronic regions (Fig. 3c, d). The GraphProt[36] motif for QKI-5 closely resembled the known QKI core motif ACUAAY[37,38],

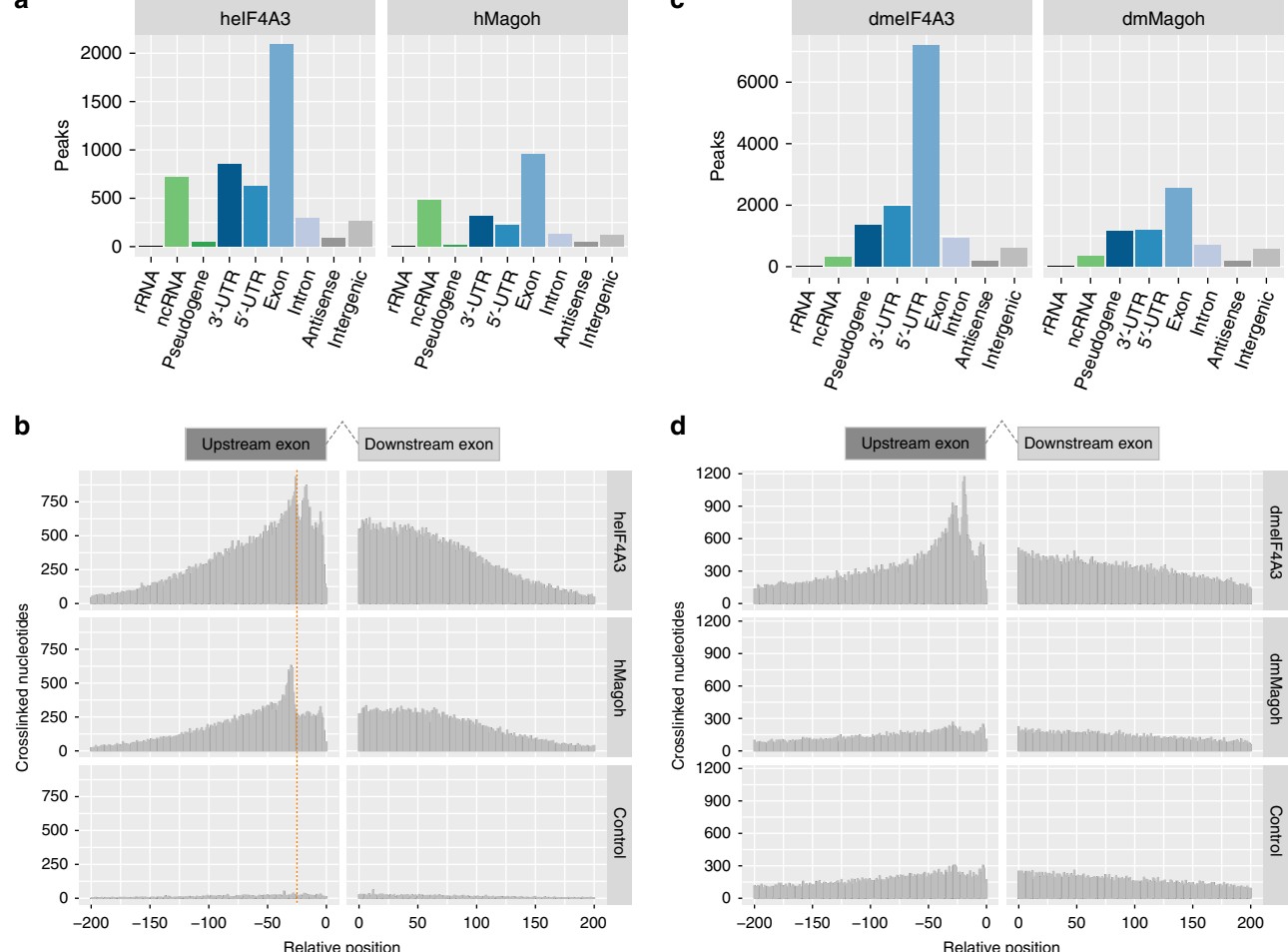

**Fig. 4** EIF4A3 binds upstream of exon–exon junctions. **a** Number of JAMM peaks located on genomic target classes for human EIF4A3 and MAGOH. Priority of target classes as in Fig. 3. **b** Histogram of crosslinked nucleotide positions for human EIF4A3, MAGOH and the corresponding control. EIF4A3 and MAGOH show increased binding 20–30 nucleotides upstream of exon–exon junctions, binding decreased with increasing distance to exon–exon junctions whereas the control is uniformly distributed. **c** Number of JAMM peaks located on genomic target classes for Drosophila EIF4A3 and MAGOH. Priority of target classes as in Fig. 3. **d** Histogram of crosslinked nucleotide positions for Drosophila EIF4A3, MAGOH and the corresponding control. EIF4A3 shows increased binding 20–30 nucleotides upstream of exon–exon junctions whereas crosslinked nucleotides of MAGOH and the control are not specifically enriched

whereas the motif for QKI-6 better resembled the consensus motif AYUAAY identified from highly expressed PAR-CLIP read clusters[8] (Fig. 3c, d).

We next compared uvCLAP QKI binding sites with sites derived from QKI PAR-CLIP[8] (retrieved from CLIPdb[10]; peaks were called by PARalyzer[39]). JAMM produces two sets of peaks: a full set of peaks containing also small peaks and peaks with few reads, and a smaller, filtered set where these peaks are removed (in the following named JAMM full and filtered, respectively). To facilitate the comparison of the peaks from the two methods, all peaks were extended to a minimal length of 41 nucleotides, where adjacent or overlapping peaks were merged.

A comparison of the overlaps between the full uvCLAP peak sets identified 7761 sites shared between QKI-5 and QKI-6 and 2685 sites shared between uvCLAP and PAR-CLIP, leaving several thousand sites exclusive to each of the three experiments (Fig. 5a and Supplementary Fig. 5a). Using the filtered uvCLAP peaks, the number of sites shared between QKI-5 and QKI-6 uvCLAP was reduced to 30.49% (2336 sites) in comparison to the full set. Using the filtered peaks also reduced the total number of uvCLAP sites to 32.37% for QKI-5 (13,530 sites) and to 34.23% for QKI-6 (12,832 sites), while the number of sites shared between uvCLAP and PAR-CLIP was reduced to 47.78% (1283 sites) in comparison to the full list of peaks. Technical differences between CLIP-Seq methods are known causes of low peak overlaps[40]. Those include differences in RNase digestion, the wavelength used for crosslinking, and preferential crosslinking to 4-thiouridine[41], as well as the use of different isoforms and differently tagged expression vectors.

To further investigate the quality of uvCLAP peaks, we determined the occurrence of the consensus motif AYUAAY (Fig. 5b). For the full sets of uvCLAP peaks, the fraction of peaks harboring the consensus motif was smaller for uvCLAP (QKI-6: 31.34%, QKI-5: 43.51%, QKI PAR-CLIP 49.37%); however, uvCLAP identified a larger number of sites harboring the consensus motif (QKI-6: 12,212, QKI-5: 19,021) compared to 4209 sites identified by PAR-CLIP. In total, uvCLAP identified 26,503 sites containing the consensus motif, of which 24,939 were exclusive to uvCLAP. Use of the filtered uvCLAP sets increased the fraction of peaks containing the consensus motif (QKI-6: +7.36%, QKI-5: +9.96%), leading to a reduction of the number of peaks containing the motif by 57.73% (QKI-6) and 60.22% (QKI-5) (Fig. 5b), but still identified 9992 peaks containing the consensus motif that were not found by PAR-CLIP. In summary, we were able to recover known QKI sequence binding preference in uvCLAP data for two different QKI isoforms. Thus, uvCLAP identified several thousand QKI binding sites harboring the known consensus motif, which were not detected by PAR-CLIP.

Next, we evaluated three KH domain-containing RBPs KHDRBS1[42], KHDRBS2[43,44], and KHDRBS3[44] (Supplementary Fig. 2a). We found that KHDRBS1, also known as Sam68, is nuclear as reported earlier[42] and interacts primarily with introns (Fig. 3e). Similar to KHDRBS1, KHDRBS3 is also nuclear and it interacts mainly with introns (Fig. 3h). On the other hand, we found that KHDRBS2 is mainly cytoplasmic and interacts with introns and 3′-UTRs (Fig. 3g). Interestingly, all three proteins recognize a similar AU-rich RNA sequence, irrespective of their subcellular localization (Fig. 3e, g, h). We then cloned KHDRBS1[Y440F], a point mutant of KHDRBS1. KHDRBS1[Y440F] was reported to disrupt a functional NLS[45] and is cytoplasmic (Fig. 3f), as has been reported before. KHDRBS1[Y440F] recognizes an AU-rich RNA sequence very similar to the wild-type KHDRBS1 as well as KHDRBS2 and KHDRBS3 but mostly at 3′-UTRs of mRNAs rather than introns (Fig. 3f). Finally, we re-cloned KHDRBS2 with a fortuitous mutation R489K that we noticed was present in a commercially available plasmid and re-

created a cell line that expresses this variant to see if it affects KHDRBS2's target preference. We found that KHDRBS2 and KHDRBS2[R489K] have highly correlated RNA-binding profiles (Figs. 1d and 2f). KHDRBS2 recognized more intronic peaks compared to KHDRBS2[R489K] (Fig. 5c). Motif analysis of KHDRBS1-3 and the two-point mutants revealed similar AU-rich motifs for all proteins (Fig. 3e–h), matching the known affinity of KHDRBS1 to UAAA[46] and U(U/A)AA repeats[38]. A preference for UAA was also shown for all three KHDRBS proteins by RNAcompete[5]. For KHDRBS1, 31.5% (19,545 sites) from the full set of peaks and 40.75% (7902 sites) from the filtered set of peaks were also found in KHDRBS1 eCLIP for K563 cells[17] (Fig. 5d and Supplementary Fig. 5b). These results show that uvCLAP protocol and the analysis pipeline described here yield robust results even when independently generated cell lines are used to probe the same or very similar proteins.

Since STAR proteins contain a KH domain, that acts as the RNA-binding module, we decided to also investigate the prototypical KH domain-containing protein hnRNPK. The triple-tagged clone of hnRNPK was predominantly nuclear (Fig. 3b). Our uvCLAP data show that hnRNPK interacts mainly with cytosine-rich RNA, specifically when RNA contains three or more consecutive cytosines (Fig. 3b). Comparison with two hnRNPK eCLIP experiments in K562 and HepG2 cells revealed that 49.74% (148,480) of uvCLAP sites were matched by sites from one of the two eCLIP experiments. For the filtered list of uvCLAP peaks, the overlap dropped to 33.23% (21,281) of uvCLAP peaks (Fig. 5d and Supplementary Fig. 5c). The GraphProt motif, determined from hnRNPK uvCLAP data (Fig. 3b), closely resembled the consensus motif determined by SELEX[47].

We found considerable numbers of common peaks when comparing to PAR-CLIP and eCLIP data, however, all comparisons also showed peaks unique to uvCLAP, PAR-CLIP, and eCLIP. This is a common observation when comparing CLIP-Seq results on the peak level[40]. Additional research will be required to pinpoint the exact contributions of CLIP-Seq method, cell line, peak calling, and lab bias when comparing CLIP-Seq experiments on the peak level. Taken together, our results show that KH domains are not restricted to AU-rich motifs (KHDRBS proteins), but can also recognize C-rich sequences (hnRNPK). Overall, the uvCLAP-derived binding sites for QKI-5, QKI-6, KHDRBS1-3, and hnRNPK matched prior knowledge regarding binding motifs and binding site localization.

## Discussion

uvCLAP allows to identify and characterize targets of RBPs in vivo without having to resort to labor-intensive techniques that use radioactive substances. Using highly stringent purification conditions, we are able to remove nonspecifically interacting RNA (Fig. 1c) and other noncovalent interaction partners (Fig. 1b). Straightforward multiplexing of experiments leads to further time savings by allowing the parallel processing of multiple samples. We show that uvCLAP works well for human, mouse, and fly cell lines.

A major benefit of joint amplification and sequencing of multiplexed uvCLAP experiments is the preservation of RNA quantities, which allowed us to directly determine the observed proportion of nonspecific background within uvCLAP libraries. For most experiments, we detected very low amounts of background. The vast majority of called peaks were not located in the vicinity of control events, emphasizing the capability of uvCLAP tandem purification to effectively remove nonspecific background. Experiments with low signal-to-control ratio and increased occurrence of control events near binding sites are

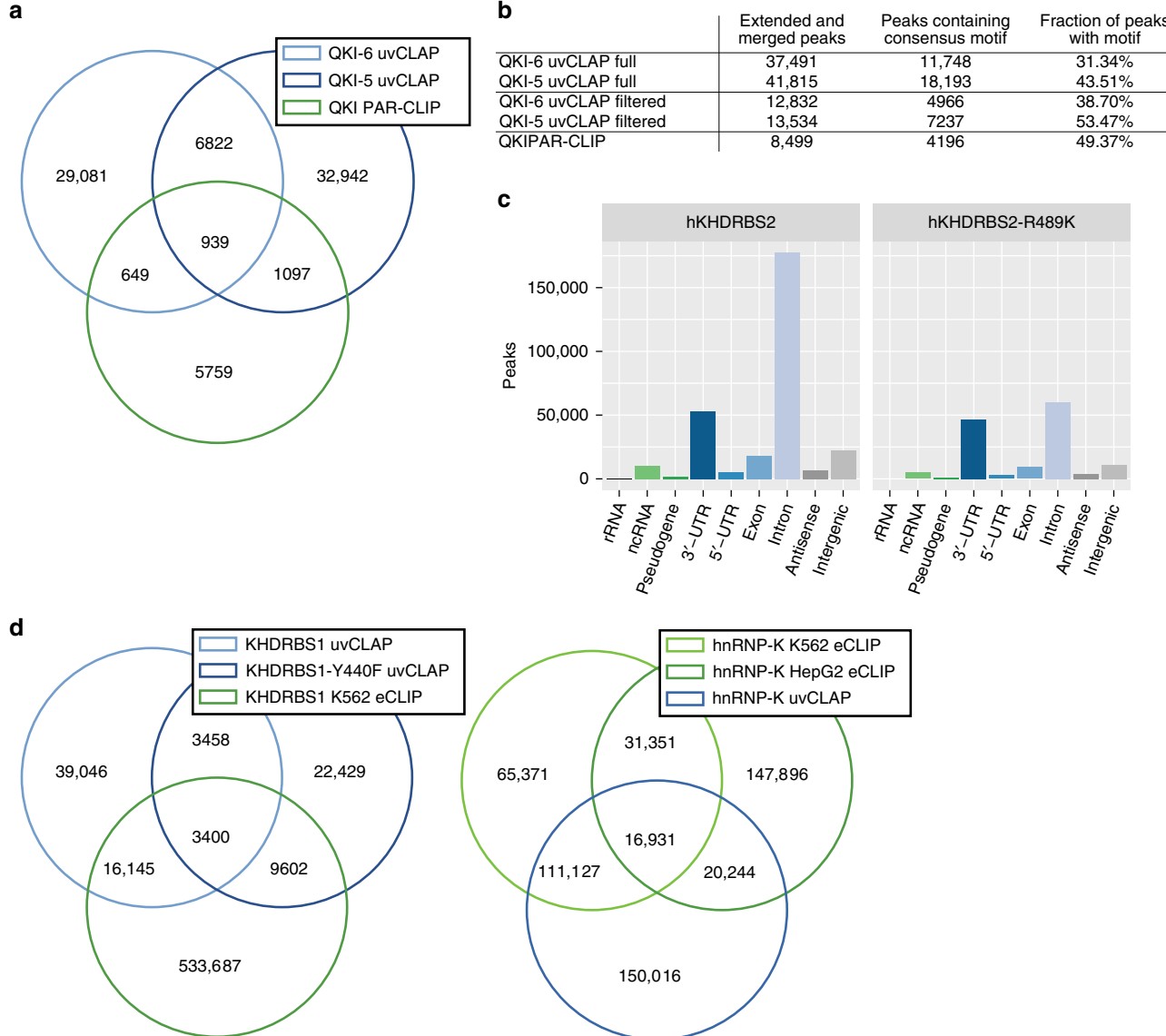

**Fig. 5** Comparison of uvCLAP to other methods. **a** Overlap between the full sets of QKI-5 and QKI-6 uvCLAP and QKI PAR-CLIP peaks. Peaks were extended to a minimum length of 41 nt, adjacent or overlapping sites were merged. **b** QKI-5, QKI-6 and QKI PAR-CLIP sites containing the consensus motif AYUAAY. **c** Number of JAMM peaks located on genomic target classes. Each peak was assigned once to the category with highest priority corresponding to the order rRNA, ncRNA, pseudogene, 3′-UTR, 5′-UTR, exon, intron, antisense, intergenic. **d** (left) Overlap between the full sets of KHDRBS1, KHDRBS1[Y440F] uvCLAP and KHDRBS1 K563 eCLIP peaks. (right) Overlap between the full sets of hnRNPK uvCLAP, hnRNPK K562 eCLIP and hnRNPK HepG2 eCLIP peaks. Peaks were extended to a minimum length of 41 nt, overlapping or abutting sites were merged

known to either inefficiently crosslink (EIF4A3), bind to very few sites (MLE), or were deliberate mutations with impaired binding (MLE-GET, MLE-K, MLE-HR, and MLE-KHR[13]).

uvCLAP accurately identifies expected targets of proteins with well-defined functions such as eIF4A1, EIF4A3 (this study), MLE[13], and DHX9[14]. We systematically identified the in vivo targets of several STAR proteins including a point mutant and a splice variant that alter the subcellular localization of the target protein and found that all recognize a similar AU-rich motif. Two isoforms of the highly related protein QKI recognize a branch-point-like sequence. Another KH domain-containing RBP, hnRNPK, recognizes a CU-rich motif that is similar to the known in vitro motif[47]. These results collectively show that the KH domain can recognize an array of related RNA motifs and that KHDRBS proteins likely regulate a common set of mRNAs through similar binding sites that can be present in introns and/or 3′-UTRs[48].

uvCLAP experiments revealed that STAR proteins bind to similar RNA motifs despite their different subcellular localizations. This ability to identify distinct RNA motifs for highly related proteins further highlights the versatility and robustness of uvCLAP. The determination of in vivo RNA–protein interactions by uvCLAP reveals the multi-faceted nature of RBPs and uncovers compartmentalization of RBPs as an additional mechanism determining RNA target specificity.

## Methods

**Cell culture and generation of stable cell lines**. Flp-In™ T-REx™ 293 cell line was purchased from Thermo Fisher Scientific (catalog no. R780-07) and is maintained with DMEM-Glutamax (Gibco 31966) and 10% fetal bovine serum (FBS). The original cell line was maintained in zeocin-containing and blasticidin-containing medium according to the manufacturer's protocol (Thermo Fisher Scientific, catalog no. R780-07) and the zeocin selection is exchanged with hygromycin upon transgene transfection for stable cell line generation. All the transgenes were cloned into pCDNA5-FRT/To (Thermo Fisher Scientific, catalog no. V6520-20) with a C-

term 3xFLAG-HBH tag and were co-transfected with pOG44 plasmid with a 1:9 DNA concentration ratio as suggested by the manufacturer's protocol. Cells were re-plated in different dilutions (1:2, 1:3, and 1:6) 24 h after the transfection and selection with 150 μg/mL hygromycin was initiated 48 h after transfection. Cell lines were maintained with blasticidin and hygromycin at all times and the transgenes were induced with with 0.1 μg/mL doxycycline for 16 h both for the uvCLAP and immunofluorescence experiments. Mouse embryonic stem cells were maintained with 15% FBS, 2000 U/mL leukemia inhibitory factor, sodium pyruvate, nonessential amino acid, and 0.1 nM β-mercaptoethanol supplemented into DMEM-Glutamax. CRISPR/Cas9 facilitated endogenous tagging of the mouse Msl1, Msl2, and Dhx9 was performed in a mouse ES cell line (WT26 male ES cell line was a kind gift of Jenuwein Lab) using CRISPR-Cas9 and single-stranded oligo donors targeting the endogenous loci[14]. Drosophila S2 cells were a gift of Prof. Matthias Hentze are maintained in Schneider's Drosophila Medium (Gibco, 11720-034) and stable cell lines are generated by transfecting the cells with expression vectors that carry a Neomycin-resistance cassette and subsquent selection with 1 mg/mL geneticin (Thermo Fisher Scientific, 10131-027) for 2 weeks[13].

**Immunofluorescence**. Doxycycline-induced cells were crosslinked with 4% methanol-free formaldehyde in phosphate-buffered saline (PBS) at room temperature for 10 min and permeabilized with 0.1% Triton-X and 1% bovine serum albumin (BSA) in PBS for 30 min at room temperature. Primary FLAG-M2 (Sigma-Aldrich, F1804) antibody was diluted (1:500) in PBS with 0.1% Triton-X and 1% BSA and incubated with fixed cells at 4 °C for ~16 h. Fluorescently labeled secondary antibodies with the appropriate serotype were used reveal target proteins. Hoechst 33342 to stain DNA. Imaging was performed with a Leica SP5 confocal microscope.

**uvCLAP procedure**. Doxycycline-induced FLPin Trex HEK293 cells are rinsed with PBS and the protein of interest is crosslinked to its cognate RNA by UV irradiation (0.15 mJ/cm$^2$ UV-C light at ~254 nm). Following crosslinking, cells are pelleted by centrifugation, snap-frozen in liquid nitrogen, and kept at −80 °C until use. Cells are then defrozen on ice and lysed with 0.5 mL of lysis buffer (FLAG immunoprecipitations: 50 mM Tris-Cl, pH 7.4; 140 mM NaCl, 1 mM EDTA, 1% Igepal CA-630, 0.1% SDS, 0.1% desoxycholate; His pulldowns: 1 × PBS, 0.3 M NaCl, 1% Triton-X, 0.1% Tween-20), mildly sonicated, and immunoprecipitated with anti-FLAG beads for 1 h at 4 °C or incubated with His-Tag Pulldown Dynabeads (10103D, Thermo Fisher Scientific) for 10 min. The beads are washed with lysis buffer and bound material is eluted with 3xFLAG peptide (250 μg/mL) or 250 mM imidazole in respective lysis buffer. The eluate is then incubated with MyONEC1 beads to collect biotinylated target protein, after which the beads are washed with high-stringency buffers (0.1% SDS, 1 M NaCl, 0.5% LiDS, 0.5 M LiCl, and 1% SDS, 0.5 M LiCl) to aggressively remove nonspecific interactors. To trim the crosslinked RNA, the beads were resuspended with 1 mL of NDB (50 mM Tris-Cl, pH 7.4; 100 mM NaCl, 0.1% Tween-20), to which 2 μL of TURBO DNase (AM2238, Thermo Fisher Scientific) and 10 μL of diluted RNaseI (1:2000–1:8000 dilution in NDB from 100 U/μL stock (AM2294, Thermo Fisher Scientific)) were added. The solution is incubated at 37 °C for 3 min and immediately transferred to a metal block cooled on ice. Dephosphorylation of cyclic phosphate groups was carried out with T4 PNK (10 U/μL, M0201, NEB) in a low pH buffer (25 mM MES (2-(N-morpholino)ethanesulfonic acid), pH 6.0; 50 mM NaCl; 10 mM MgCl$_2$; 0.1% Tween-20; 20 U RNasin (N2511, Promega); 10 U PNK; 20 min at 37 °C). 3′-Linkers are then ligated with T4 RNA ligase 1, excess adapters are washed away, and 3′-tagged, crosslinked RNA is released with proteinase K digestion and column purification (Zymo DNA Clean and Concentrator). Reverse transcription is carried out with SuperScript III and barcoded reverse transcription primers. After reverse transcription, relevant samples are mixed and the cDNA is separated on a 6% 6 M urea PAA gel. Size-fractionated cDNA is then circularized with CircLigase, linearized by restriction digestion, and amplified by PCR to generate sequencing libraries.

In order to minimize nonspecific contaminants that arise from abundant RNA species such as ribosomal RNA, small nuclear RNA, and transfer RNA, uvCLAP relies on in vivo target protein biotinylation and a quick tandem affinity purification under very stringent conditions. We achieved biotinylation either by exogenously expressing proteins of interest with a biotinylatable peptide or via introduction of the tag into the endogenous locus by CRISRP/Cas9. Due to this dependence on biotinylation, uvCLAP cannot be used when these methods are not feasible (e.g., primary cells) or when affinity-tagging changes target protein localization and/or activity.

**Silver gel staining**. Silver gel staining (Fig. 1b) was performed using the Silver Quest Silver Staining Kit (Thermo Fisher LC6070) as per the manufacturer's instructions.

**RNA visualization**. In order to evaluate purified RNA in a uvCLAP experiment (Fig. 1c and Supplementary Fig. 1b), following purification from UV-crosslinked or non-crosslinked cells, total RNA was isolated from FLAG, or streptavidin beads with proteinase K and purified with Zymo DNA Clean and Concentrator columns. Purified RNA was loaded on a Agilent RNA 6000 Pico chip and analyzed using

Bioanalyzer 2100 as per the manufacturer's recommendations. Uncropped images are available in Supplementary Fig. 6.

**uvCLAP tri-barcode approach**. The uvCLAP tri-barcode approach—based on ScriptSeq PCR primers and two custom 5 nucleotide barcodes adjoining the 5′-adapter and 3′-adapter—allows to flexibly tag source libraries according to multiple experimental conditions. This tagging strategy enabled us to tag pulldown conditions, size fractions, and biological replicates prior to PCR amplification and sequencing. Custom 5′-tags and 3′-tags were used to distinguish pulldown condition and biological replicates. Size fractions were differentiated using commercially available ScriptSeq PCR primers.

ScriptSeq primer sequences were: AATGATACGGCGACCACCGAGATCTACACTC TTTCCCTACACGACGCTCTTCCGATCT (5′ adapter) and CAAGCAGAAGACGGCATACGAGATNNNNN NGTGACTGGAGTTCAGACGTGTGCTCTTCCGATCT (3′ adapter, "NNNNNN" shows the position of the specific PCR index).

To ensure optimal in silico separability of experimental conditions, we used edittag[50] to design tags robust to indel (insertion/deletion) and substitution errors. For this purpose, we created a set of five nucleotide tags with minimal pairwise edit distance of 3, ensuring that 1 indel or substitution error can be corrected. The initial set of 31 candidate tags, created so as not to contain polybases or self-complements, was further filtered for tags containing nucleotide repetitions at either end (11 tags) and tags reverse complementary to the adapter (5 tags), leaving 15 tags for use by uvCLAP (Supplementary Table 7). The combination of tags used for each multiplexed library was chosen to provide at least one nucleotide detected by the red and green color channels used by Illumina sequencers at each position.

To distinguish pairwise biological replicates, we created semi-random tags according to patterns DRYYR and DYRRY (IUPAC ambiguity code, D: not C, R: purine, Y: pyrimidine). Any pair of tags created according to these patterns has a guaranteed minimum edit distance of 2; correspondingly, these tags are not error correcting in the context of indel and substitution errors. These tags are robust to substitution errors as are predominantly produced by Illumina-type sequencing, requiring four substitutions to erroneously assign any given tag to the wrong pattern.

uvCLAP uses random barcodes that serve as UMIs to identify individual crosslinking events from sets of potentially large PCR duplicates. Here, edittag-designed 5′-tags were interleaved with five random nucleotides according to the pattern NNNT$_1$T$_2$T$_3$T$_4$T$_5$NN (N: random nt, T: tag nt), yielding 1024 different combinations. The semi-random barcodes positioned adjacent to the 3′-adapter provide additional 48 different combinations. In combination, these serve to extend the detectable number of crosslinking events per nucleotide position to at least 49,152 events. To reliably detect the barcodes positioned at both ends of the genomic inserts, uvCLAP by default uses paired-end sequencing. This also allows the use of the genomic positions of both insert ends during PCR duplicate removal to further increase the number of detectable crosslinking events, depending on the number of different insert lengths in the sequenced library.

**Joint amplification of libraries preserves RNA quantities**. Each multiplexed uvCLAP contains at least two biological replicates of signal and control libraries. Joint amplification and sequencing of these experiments preserves RNA quantities between the multiplexed libraries both at the global and at the binding site level, allowing their direct comparison. While methods that necessitate library normalization only allow the comparison of conditions that share a large number of binding sites of similar strengths, uvCLAP preserves the quantitative relationship of samples with identical pulldown conditions; the proportions of observed events correspond to the proportions of RNA in the samples. Hence, a direct comparison between conditions can be performed whenever the comparison of the RNA amounts would be meaningful, that is, when the target RBPs are normally expressed and when the pulldown efficiency of the targeted RBPs is similar. For this purpose, we suggest using endogenously tagged proteins for the comparison of specific pulldown conditions. Preservation of relative RNA quantities makes uvCLAP ideally suited for the investigation of differential binding of multiple protein isoforms or knockdown conditions. The straightforward comparability of arbitrary pulldown conditions also allows for a detailed investigation of altered binding exhibited by RBP deficiency mutants. Importantly, these controls are also effective when there is no prior knowledge of the expected binding behavior.

**uvCLAP data processing**. uvCLAP libraries were demultiplexed and adapters removed using Flexbar (version 2.32)[51]. Barcodes and UMIs were extracted using custom scripts. To reliably remove readthroughs into barcode regions containing random and semi-random nucleotides, five nucleotides (corresponding to the length of semi-random 3′ tags) were clipped from the 3′ ends of first mate reads, 10 nucleotides (corresponding to the length of 5′ tags and UMIs) were clipped from 3′ ends of second mate reads. Since any genomic sequence removed by this step is guaranteed to be contained in the other mate, no information is lost. Bowtie2 (version 2.2.2)[52] was used to map demultiplexed and processed reads to reference genomes hg19, dm3, and mm10. Uniquely mapped reads were extracted by removing all reads for which multiple alignments could be identified by bowtie2 as

indicated by the "XS:i" SAM flag. Alignments sharing UMIs and start coordinates of first and second mate reads were combined into individual crosslinking events. We removed all crosslinking events supported by less than 10% of the reads compared to the event with the highest number of reads at the same genomic position as they were most likely spurious events arising from errors introduced into UMIs during amplification or sequencing[22].

HISAT2[31,32] alignments were performed using version 2.0.5 and "parameters --fr --no-mixed --no-discordant".

Peak calling refers to the identification of discrete binding sites from reads or binding events and usually includes an additional filtering to reduce the number of false-positive binding sites. However, we found most strategies such as the presence of specific mutations caused by crosslinked nucleotides[39,53], enrichment over shuffled input signal[54–56], or modeling of read-count distributions[57,58] not suitable because of the specifics of our data.

The majority of peak calling methods compare the number of crosslinking events of a region to a baseline distribution in order to determine regions with a significantly enriched number of events. There are in general two principal approaches for determining this baseline. The first approach, namely shuffling of the gene-wise input signal, assumes a uniform distribution for the baseline. However, this uniformity is neither observed with uvCLAP nor the PAR-CLIP nonspecific background[23]. The second approach uses global read-count distributions and assumes that most reads are nonspecific. This assumption is again not indicated by our data. A third possibility would be to omit any filtering after the initial peak identification presuming that our libraries are mostly free of background[6,7]. This was also rejected as there are varying numbers of nonspecific events seen in our data.

Thus, we looked for a peak calling method that is able to incorporate quantitative controls into the peak calling procedure. This requirement is satisfied by JAMM[26], a universal peakfinder that is able to integrate information from uvCLAP biological replicates and background controls, and PEAKachu[27] a stringent peak caller that utilizes the rigorous statistical evaluation of DESeq2[59]. JAMM typically reports a large number of peaks, allowing for a flexible posterior filtering.

Peaks were called using JAMM[26] (version 1.0.7rev1, parameters "-d y -t paired -b 50 -w 1 -m normal")[26] based on crosslinked nucleotides of biological replicates of signal and control libraries. An additional set of peaks was called using PEAKachu[27] (version 0.0.1alpha2, https://github.com/tbischler/PEAKachu/ releases/tag/0.0.1alpha2, parameters --pairwise_replicates --max_proc 3 --max_insert_size 100 -m 0 -n manual --size_factors 1 1 0.75 0.75).

Pairwise Spearman's correlations for Fig. 2f were calculated using deeptools[60] (version 2.3.5)[60] based on the number of events on the merged peak regions of the full sets of JAMM peaks. Spearman's correlations of biological replicates based on PEAKachu peaks were calculated using bedtools[61], parallel[62] and R (version 3.4.2).

GraphProt[36] sequence models were trained on 10,000 randomly selected peaks and roughly equal numbers of unbound sequences, using default parameters (GraphProt version 1.1.3)[36]. Unbound sequences were selected by randomly placing peaks within genes with at least one binding site and at least 100 nucleotides apart from any bound site. For training and motif generation, the 60 nucleotides surrounding peak centers were used. Motifs were generated based on the 5% highest-scoring peaks among the 10,000 bound training instances.

Venn diagrams comparing uvCLAP, PAR-CLIP, and eCLIP peaks were created using pybedtools[63] (version 0.7.9)[63].

**Antibodies used for immunoblots.** FLAG-HRP (Sigma-Aldrich, A8592) used at 1:2000 dilution in Fig. 1c, Supplementary Fig. 1b, and Supplementary Fig. 1c. Sam68 (KHRDBS1) polyclonal antibody (Sigma-Aldrich, S9575) used at 1:1000 dilution in Supplementary Fig. 1c. Tubulin monoclonal antibody (Sigma-Aldrich, T9026) used at 1:5000 dilution in Supplementary Fig. 1c.

**Data availability**. The uvCLAP data in this study has been deposited to the Gene Expression Omnibus database under the accession numbers GSE87792 (https:// www.ncbi.nlm.nih.gov/geo/query/acc.cgi?acc=GSE87792) (MLE) and GSE85155 (https://www.ncbi.nlm.nih.gov/geo/query/acc.cgi?acc=GSE85155) (all other proteins). PEAKachu peaks are available at Zenodo (https://zenodo.org/record/ 1063948)[64].

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

## Acknowledgements

We thank F. Heyl, U. Bönisch, and the members of the Deep Sequencing Facility of the Max Planck Institute of Immunobiology and Epigenetics. We extensively used software distributed via bioconda, a comprehensive software distribution for the life sciences[49]. This work was supported by CRC992 (A.A., R.B.), CRC746 (A.A.), and CRC1140 (A.A.).

## Author contributions

D.M., I.A.I., and T.A. designed the experiments; I.A.I. and T.A. performed the experiments; I.A.I., T.A., and D.M. developed the uvCLAP method with input from R.B. and A.A.; D.M. analyzed uvCLAP data and was supervised by R.B.; D.M., I.A.I., T.A., R.B., and A.A. wrote the manuscript; R.B. and A.A. acquired funding; A.A. and R.B. supervised all aspects of the study. All authors reviewed, edited, and approved the paper.

## Additional information

**Competing interests:** The authors declare no competing interests.

