## [Peer Review File(PDF 321 kb) · Nature Communications]

Reviewers' comments:

Reviewer #1 (Remarks to the Author):

First, my apology for the delay.

This is a very well written article describing, what appears to be an excellent new method for identifying in vivo targets of RBPs and their binding motifs. uvCLAP also appears to represent a considerable technical improvement over traditional CLIP based methods. I have no specific concerns or issues but do have some general comments that may be worth considering.

1. In general, for this type of method the use of three replicates is preferred to duplicates, which also permits a more robust statistical analysis of the data.
2. The use of the terms "foreground" and "background" throughout the text seemed odd. If this is referring to signal and noise, these terms may be better alternatives.
3. A comparison to PAR-CLIP would have been helpful.
4. The relatively limited overlap with compared CLIP data is a bit troubling.
5. Figure 2a is not very helpful and difficult to completely understand. For instance, why is light blue section depicting the 4th step smaller in width?
6. Although a minor point, the term "clap" has a relatively negative connotation, at least in the US, and may not be an ideal choice name for a method (see <http://www.urbandictionary.com/define.php?term=the%20%22clap%22>)

Reviewer #2 (Remarks to the Author):

Maticzka, Ilik, Aktas et al. developed a novel method to identify targets of RNA binding proteins (RBP), called UV-CLAP. At the difference of CLIP approach, this technic allows the use of very stringent conditions to purify crosslinked RNA-RBP complex and does not require radioactive labeling of RNAs. Also, the authors use a barcode system to label and combine many samples in a single experiment. While these aspects can represent an advantage, the present uvCLAP method is poorly controlled and the easiness of the protocol appears modestly improved when compared to CLIP approach.

Specific points

- 1) The specificity and reproducibility of the uvCLAP signals are major concerns.
 - a) While very stringent conditions were used to purify RNAs bound to the tagged-RBP, the authors should ensure that all free RNA were eliminated at the end of the pull-down. Radiolabelling of RNA and analysis on denaturing gel analysis could be done in one experiment as proof of principle of the stringency of the purification.
 - b) Why is the background signal so high in drosophila sample (10 times more crosslink events in drosophila than human and mouse, Supplementary Figure 1c-e)?
 - c) The correlation of crosslinked event number located on merged peaks for biological replicates appears low (in average 0.46 correlation, Figure 2c). Notably, from figure 2d, it appears that regions covered by a low and middle number of crosslinked events exhibit the highest variability between biological replicates. Did the authors try to use more stringent filters to only select robust peaks covered by a significant number of crosslinked events?

- 2) The approach used to quantify amount of crosslinked events is not convincing.
- a) The authors choose to not add any normalization step and keep absolute number of crosslinked events. This is directly linked to the level of tagged-protein present in the sample. The authors have to control that uvCLAP is initially performed with equivalent level of tagged-protein. This is very important for reproducibility between biological replicates and comparison between foreground-to-background.
 - b) Did the author check whether the barcode used lead to any sequencing bias and consequently error in quantification? It is well recognized that some sequences can introduce bias during amplification and sequencing steps.
- 3) The validity of uvCLAP method was assessed by comparing targets identified in previous published CLIP data (figure 5 and supplementary figure 4). Comparison of both technics reveals quite different results. Since CLIP and CLAP experiments were performed in different systems (different cell lines, different antibodies...), the interpretation of these differences is difficult. The authors have to compare uvCLAP and CLIP methods in a single experiment performed in parallel.
- 4) The expression of the tagged-RBP results in overexpression of the RBP (except for mmDHX9, mmMSL1, mmMSL2 for which endogenous genes were tagged; but note that MSL1 and 2 exhibit low foreground-to-background ratios, Figure 2f). Given that many RBPs act in a dose-dependent manner this is a concern about the validity of the target sites extracted from the data.
- 5) Why the authors did not use any RNase treatment step? In principle, use of RNase after the pull-down increases the resolution of RBP-binding sites. Consistent with a modest resolution of uvCLAP, there is not a clear and distinct eIF4A3-peak 20-30nt upstream the exon-exon junction (EEJ), but rather a slight and spread peak gradually decreasing with distance from EEJ. (Figures 4c,d)
- 6) In the end, the advantages brought by uvCLAP in comparison to CLIP seems modest. uvCLAP doesn't require radioactive labeling of RNA (if validated, cf point 1) but still require size fractionation through denaturing gel step (cf figure 2e and supplementary 3 highlighting size fragment-bias).
- 7) Overall, the manuscript is quite difficult to read. Many technical precisions in the text are missing (i.e. for the control experiments, which ones correspond to the use of tagged-GFP and those to the use of 3FHBH-tag itself? ; why the authors choose UV-C for crosslink?). Also some figures are not properly annotated or not mentioned at all in the text and many of them don't appear in the proper order in the text.

Manuscript: NCOMMS-17-24436

Point by point response to reviews

Reviewer #1:

This is a very well written article describing, what appears to be an excellent new method for identifying in vivo targets of RBPs and their binding motifs. uvCLAP also appears to represent a considerable technical improvement over traditional CLIP based methods.

We thank the reviewer for his/her supportive comments and constructive suggestions to improve the manuscript.

I have no specific concerns or issues but do have some general comments that may be worth considering.

1. In general, for this type of method the use of three replicates is preferred to duplicates, which also permits a more robust statistical analysis of the data.

We used two replicates as a compromise between investigating as many different samples as possible while maintaining sufficient statistical power. We are working on ways to simplify the protocol further that would make easier to use 3 or more replicates per sample. We would like to note that, although this was not the intention, we ended up using 4 replicates for KHDRBS2 since the point mutant behaves almost identical to the wild-type construct (see Fig. 1c and Fig. 2c).

2. The use of the terms "foreground" and "background" throughout the text seemed odd. If this is referring to signal and noise, these terms may be better alternatives.

We now use signal and control instead of "foreground" and "background" to be more clear and only use the term background control where necessary to distinguish between other types of controls, i.e. input controls.

3. A comparison to PAR-CLIP would have been helpful.

We have now added this comparison, see Supplementary Figure 1.

4. The relatively limited overlap with compared CLIP data is a bit troubling.

We agree that at first sight this could indeed be troubling. However, there are various reasons why this is the case: The use of different cell lines might be a big factor, however, there is also the issue of sampling a very large interaction space with the very specific but very inefficient UV-crosslinking method. Combining these factors with very long, >100 step CLIP protocols with multiple gel extractions on very little starting material lead inevitably to a lot of variability, even within the same experiment. For example, if one looks at ¹, the word “reproducibility” does not exist in the entire manuscript, including the supplementary material.

Importantly however, what we do show is internal consistency and reproducibility, not just with replicates, but with mutants and proteins of the same/similar family. This is very nicely demonstrated in our Fig. 2c, where we show an unsupervised clustering of uvCLAP peaks for all of the proteins we have surveyed in human cells. Not only do replicates agree very well with each other, but KHDRBS proteins cluster together (also note the clustering of KHDRBS2 and KHDRBS2^{R489K} samples, which are prepared independently and processed months apart from each other). We can also see that the two isoforms of QKI (QKI-5 and QKI-6) clusters with each other, moreover, the cytoplasmic isoform (QKI-6) is closer to the cytoplasmic KHDRBS proteins we used (KHDRBS2, KHDRBS2^{R489K}, KHDRBS^{Y440F}) than the nuclear isoform, QKI-5. Finally the EJC components Magoh and eIF4A3 cluster separately with each other and with eIF4A1 which is a close homologue of eIF4A3, and hnRNP-K, with its C-rich binding sequence is further away from all the STAR proteins which prefer A/U-rich sequences (also compare with Fig. 3).

In summary, we show that using our method produces internally consistent and reproducible results, but if one wants to be absolutely certain about a binding site in another system at a particular site in the genome, it is advisable to carry out the experiment in that system and use orthologues methods to verify that binding site. (see p. 10, highlighted in blue)

5. Figure 2a is not very helpful and difficult to completely understand. For instance, why is light blue section depicting the 4th step smaller in width?

We have now expanded the figure legend for this figure to make it more clear.

6. Although a minor point, the term "clap" has a relatively negative connotation, at least in the US, and may not be an ideal choice name for a method (see <http://www.urbandictionary.com/define.php?term=the%20%22clap%22>)

We were indeed not aware of this connotation, and thank the referee for pointing out. It is unfortunately a little bit late to change the name since we have reported the use of the method in two publications now: ^{2,3}. Moreover, the term we use for the method is uvCLAP and not CLAP alone.

Reviewer #2:

Maticzka, Ilik, Aktas et al. developed a novel method to identify targets of RNA binding proteins (RBP), called UV-CLAP. At the difference of CLIP approach, this technic allows the use of very stringent conditions to purify crosslinked RNA-RBP complex and does not require radioactive labeling of RNAs. Also, the authors use a barcode system to label and combine many samples in a single experiment. While these aspects can represent an advantage, the present uvCLAP method is poorly controlled and the easiness of the protocol appears modestly improved when compared to CLIP approach.

Specific points

1) The specificity and reproducibility of the uvCLAP signals are major concerns.

a) While very stringent conditions were used to purify RNAs bound to the tagged-RBP, the authors should ensure that all free RNA were eliminated at the end of the pull-down.

While we agree with this opinion in principle, in practice, removing all free RNA is simply impossible, and we think that this idea has been systematically debunked for PAR-CLIP already^{4,5} and there is no reason to doubt that similar problems exist in other protocols as well. This is precisely the reason why we spent time developing and optimizing this protocol, so that we can use negative controls (only-tag, or GFP-tag expressing cell lines) in parallel with all of experiments to detect, quantify and eliminate the background, which includes, among other things, free RNA.

The elimination of free RNA at the end of the pull-down is a fundamental prerequisite for the specificity of CLIP-seq methods. The overall goal, however, is having low amounts of background included after sequencing. And this depends on the amount of RNA in the signal library. The results of Friedersdorf and Keene indicate as much, observing varying amounts of overlap with background based on a single independent control experiment and varying only the experiments that it was compared to.

As a consequence of this, we decided to always include matched background controls for all experiments to control for the amounts of actually observed background.

Radiolabelling of RNA and analysis on denaturing gel analysis could be done in one experiment as proof of principle of the stringency of the purification.

We have done this in a separate manuscript that deals with the biology of the RNA helicase MLE².

b) Why is the background signal so high in drosophila sample (10 times more crosslink events in drosophila than human and mouse, Supplementary Figure 1c-e)?

Thank you for highlighting this important topic. This was not discussed in the manuscript and we now have added a detailed discussion of this effect.

In short, this is caused by the small size of signal libraries that were multiplexed with this control. The control containing the large number of events was multiplexed with experiments for dmEIF4A3 (barely enriched over the control) and dmMagoh (not enriched) (library L1). Hence, the levels of RNA pulled down are comparable to those of the control.

A large number of events is however not necessarily caused by large amounts of RNA in a library. In this case, sufficiently strong amplification and deep sequencing combined with the lack of competition against real binding events enabled the sequencing of a much larger number of control reads compared to other libraries. This effect was also exemplified by the PAR-CLIP controls that sequenced 3 times 180 million reads of background which allowed the detection of very large numbers of sites. This effect is inherent to current high-throughput sequencing which is why the direct comparison of signal and background as done for uvCLAP is required to put these numbers in context. (see p. 8, highlighted in blue)

c) The correlation of crosslinked event number located on merged peaks for biological replicates appears low (in average 0.46 correlation, Figure 2c). Notably, from figure 2d, it appears that regions covered by a low and middle number of crosslinked events exhibit the highest variability between biological replicates. Did the authors try to use more stringent filters to only select robust peaks covered by a significant number of crosslinked events?

The correlation heatmap in Figure 2c globally compares all human uvCLAP experiments and is meant to give a global overview and to show that the measurements for the different pulldown conditions differ. To enable this we merged all peak regions and calculated correlations on a total of ~800.000 regions. On the basis of such a large number of data points, an average correlation of 0.46 is actually quite good.

We have added the additional information necessary to put this into proper context and now also report correlations between biological replicates using a more stringent set of peaks, resulting in an average correlation of 0.92 between biological replicates. (see p. 10, highlighted in blue)

2) The approach used to quantify amount of crosslinked events is not convincing.

a) The authors choose to not add any normalization step and keep absolute number of crosslinked events. This is directly linked to the level of tagged-protein present in the sample. The authors have to control that uvCLAP is initially performed with equivalent level of tagged-protein. This is very important for reproducibility between biological replicates and comparison between foreground-to-background.

The main aim of library normalization is to adjust for differences in library preparation and sequencing. With uvCLAP we have matched amplification and sequencing of the samples that should be compared, namely the biological replicates of signal and control experiments. Nonetheless, differences between experiments expected to influence the total number of events remain. The remaining differences between multiplexed uvCLAP experiments are the expression levels of tagged proteins and the barcodes.

While the levels of tagged protein present in the samples are one of the principal determinants on the number of binding events, the number of observed events is influenced by many additional factors up-to and including the bioinformatics analysis. For this reason we found it more vital to evaluate the overall end-product. Our analysis of the observed events shows that the numbers of detected events between biological replicates correspond very well, indicating that differing levels of expressed proteins are not an issue.

We think that the levels of pure tags or tagged GFP in the control experiments are less important for the number of detected control events, instead the counts of control events seem mostly determined by the size of the signal libraries the controls compete with for sequencing. We agree, that determining the levels of expressed proteins should be one of the prerequisites for comparing experiments targeting different proteins. While of outstanding interest, these comparisons are outside of the scope of this article and have revised the manuscript to clarify this. We hope that our current work will be a good foundation for enabling these advanced comparisons in the future.

b) Did the author check whether the barcode used lead to any sequencing bias and consequently error in quantification? It is well recognized that some sequences can introduce bias during amplification and sequencing steps.

Semi-random barcodes distinguishing biological replicates have the same sequence composition, accordingly their use might introduce additional noise but no bias. Here, the good agreement between event counts of replicates shows that these barcodes don't cause sequencing or amplification bias. The UMIs used to distinguish individual binding events are shared by all multiplexed experiments and also might introduce noise but no bias.

Bias caused by regular barcodes can be an issue when comparing different pulldown conditions tagged with different barcodes. We have taken care to mitigate potential barcode bias by interleaving the barcodes used for distinguishing pulldown with the UMIs.

In our recently published manuscript [Ilik, I. A. *et al.* A mutually exclusive stem-loop arrangement in roX2 RNA is essential for X-chromosome regulation in *Drosophila*. *Genes Dev.* **31**, 1973–1987 (2017).], we used the 18S ribosomal RNA pseudogene CR41602 that frequently turns up in *Drosophila*-based CLIP-Seq experiments as a second factor for investigating bias. The comparison of event counts for CR41602 revealed a relative standard deviation of only 12,10% across the six uvCLAP libraries, indicating that uvCLAP barcodes -- if at all --- only introduce a very mild bias. We have updated the uvCLAP manuscript to also include these findings. (see p. 7, highlighted in blue)

3) The validity of uvCLAP method was assessed by comparing targets identified in previous published CLIP data (figure 5 and supplementary figure 4). Comparison of both technics reveals quite different results. Since CLIP and CLAP experiments were performed in different systems (different cell lines, different antibodies...), the interpretation of these differences is difficult. The authors have to compare uvCLAP and CLIP methods in a single experiment performed in parallel.

It is indeed ambitious to look for reproducibility between CLIP profiles across different cell lines, as we have done in Figure 5 and Supplementary Figure 4. This notwithstanding, evaluations of this kind can serve as basis for further discussion of the comparability between CLIP-Seq experiments. Despite their differences there are important similarities, notably the motifs and types of bound targets, that appear independent of cell line, antibody or CLIP-Seq protocol. We have revised the manuscript to better highlight the similarities and discuss the results on the overlap of genomic intervals in this context.

4) The expression of the tagged-RBP results in overexpression of the RBP (except for mmDHX9, mmMSL1, mmMSL2 for which endogenous genes were tagged; but note that MSL1 and 2 exhibit low foreground-to-background ratios, Figure 2f). Given that many RBPs act in a dose-dependent manner this is a concern about the validity of the target sites extracted from the data.

We agree that overexpression is a valid concern but find that the effects of overexpression are rather an issue of physiological relevance than of validity. The same can be said regarding the use of immortalized cell lines. The conditions under which an experiment is performed must of course always be considered when evaluating the extracted data. We have extended the manuscript to provide a more detailed assessment of the benefits and limitations arising from these features of uvCLAP. We are actively working on methods that are not dependent on endogenous genes or immortalized cell lines, but this is outside the scope of this work.

5) Why the authors did not use any RNase treatment step? In principle, use of RNase after the pull-down increases the resolution of RBP-binding sites. Consistent with a modest resolution of uvCLAP, there is not a clear and distinct eIF4A3-peak 20-30nt upstream the exon-exon junction (EEJ), but rather a slight and spread peak gradually decreasing with distance from EEJ. (Figures 4c,d)

We apologise for the omission of the RNase treatment step from the Figure 1a, which describes the overall methods and the detailed explanation of the method, and thank the referee for pointing this out. We use RNase1 to trim the bound RNA both to increase the resolution of our data, similar to other CLIP protocols.

Conventional wisdom with regards to eIF4A3 does suggest that eIF4A3 signal should be more concentrated at -20-30nt before the exon-exon junctions, however either due to physiological reasons i.e. eIF4A3 positioning is simply more stochastic than predicted or due

to technical issues with UV-crosslinking i.e. the difficulty of crosslinking proteins like eIF4A3 which seem to interact almost exclusively with the phosphate backbone, neither we nor others see such sharp enrichments. For further discussion please see ⁶⁻⁸.

6) In the end, the advantages brought by uvCLAP in comparison to CLIP seems modest. uvCLAP doesn't require radioactive labeling of RNA (if validated, cf point 1) but still require size fractionation through denaturing gel step (cf figure 2e and supplementary 3 highlighting size fragment-bias).

There are many advantages to uvCLAP as compared to "normal" CLIP protocols. Not requiring radioactive labeling is an important advantage, but the ability to use a characterized affinity tag combination and purification conditions, in addition to the several RNA-binding protein profiles and profiles we have generated from negative controls from three different model organisms (*Drosophila melanogaster*, mice and human cells) we provide in this manuscript will probably not be perceived as modest by experimenters who would like to investigate multiple RNA-binding proteins, or perhaps more importantly multiple mutants of a single RNA-binding protein.

7) Overall, the manuscript is quite difficult to read. Many technical precisions in the text are missing (i.e. for the control experiments, which ones correspond to the use of tagged-GFP and those to the use of 3FHBH-tag itself? ; why the authors choose UV-C for crosslink?). Also some figures are not properly annotated or not mentioned at all in the text and many of them don't appear in the proper order in the text.

We apologize for the disordered and missing figures, we inadvertently added a previous version of the figures during submission. We have carefully checked for consistency, considerably revised the manuscript for readability and added clarifications for the above points.

References

1. Hafner, M. *et al.* Transcriptome-wide identification of RNA-binding protein and microRNA target sites by PAR-CLIP. *Cell* **141**, 129–141 (2010).
2. Ilik, I. A. *et al.* A mutually exclusive stem–loop arrangement in roX2 RNA is essential for X-chromosome regulation in *Drosophila*. *Genes Dev.* (2017).
doi:10.1101/gad.304600.117
3. Aktaş, T. *et al.* DHX9 suppresses RNA processing defects originating from the Alu invasion of the human genome. *Nature* **544**, 115–119 (2017).

4. Friedersdorf, M. B. & Keene, J. D. Advancing the functional utility of PAR-CLIP by quantifying background binding to mRNAs and lncRNAs. *Genome Biol.* **15**, R2 (2014).
5. Reyes-Herrera, P. H., Speck-Hernandez, C. A., Sierra, C. A. & Herrera, S. BackCLIP: a tool to identify common background presence in PAR-CLIP datasets. *Bioinformatics* **31**, 3703–3705 (2015).
6. Saulière, J. *et al.* CLIP-seq of eIF4AIII reveals transcriptome-wide mapping of the human exon junction complex. *Nat. Struct. Mol. Biol.* **19**, 1124–1131 (2012).
7. Singh, G., Ricci, E. P. & Moore, M. J. RIPiT-Seq: a high-throughput approach for footprinting RNA:protein complexes. *Methods* **65**, 320–332 (2014).
8. Haberman, N. *et al.* Insights into the design and interpretation of iCLIP experiments. *Genome Biol.* **18**, 7 (2017).

Reviewers' comments:

Reviewer #2 (Remarks to the Author):

Overall, the authors adequately addressed my comments. Few remaining points are listed below. If the authors address these points, I will support the publication of this work in Nature Communications.

Related to point 1) a) : One of the strength of the uvCLAP method is the stringent tandem affinity purification of the crosslinked RNA-RBP complex: this should allow a better elimination of free-RNA than with traditional IP relying on antibodies ; and then allow the experimenters to skip the step consisting in cutting out RNA-RBP complex migrating at discrete positions on denaturing gel.

As a method paper, I think this is essential to actually assess once a proof of principal the efficiency of free-RNA elimination (while I agree complete elimination is obviously impossible) by migrating the eluate of the purification on a denaturing gel (before tagging the 3'end of putative-bound RNA and their release from RBP with proteinase K treatment). This should be included in this paper (btw I have not found this experiment in the paper mentioned by the authors).

Related to point 1) c) : The very high correlation of peaks between biological replicates revealed with the use of the stringent peak calling method PEAKachu is nicely convincing. In addition, giving the correlation values between unrelated replicates – as done with JAMM peakfinder in figure 2c- would even strengthen this analysis.

Related to point 7) : Some figures still appear in a different order in the text versus the figure panels (see figure 2)

Manuscript: NCOMMS-17-24436 / Point by point response:

Overall, the authors adequately addressed my comments. Few remaining points are listed below. If the authors address these points, I will support the publication of this work in Nature Communications.

Related to point 1) a) : One of the strength of the uvCLAP method is the stringent tandem affinity purification of the crosslinked RNA-RBP complex: this should allow a better elimination of free-RNA than with traditional IP relying on antibodies ; and then allow the experimenters to skip the step consisting in cutting out RNA-RBP complex migrating at discrete positions on denaturing gel.

As a method paper, I think this is essential to actually assess once a proof of principal the efficiency of free-RNA elimination (while I agree complete elimination is obviously impossible) by migrating the eluate of the purification on a denaturing gel (before tagging the 3'end of putative-bound RNA and their release from RBP with proteinase K treatment). This should be included in this paper (btw I have not found this experiment in the paper mentioned by the authors).

We would like to thank the reviewer for this suggestion. We performed the requested experiment and added it as Fig. 1c and Supplementary Fig. 1b.

In summary we carried out purifications using cell lines that express either KHDRBS2^{3FHBH} or GFP^{3HBH}, either with UV-crosslinking or no crosslinking and either only via beads coupled to FLAG antibody or using our stringent purification scheme (polyhistidine, followed by streptavidin pull-down with up to 1% SDS). Instead of going through with adapter ligation and proceeding with the rest of the cloning procedure, we isolated RNA from FLAG or streptavidin beads using proteinase K and analyzed the eluates on a Bioanalyzer RNA chip. Supporting our observations from sequencing data, we observed a smear of RNA specifically in the UV-crosslinked KHDRBS2^{3FHBH} but not with non-crosslinked sample (Fig. 1c and Supplementary Fig. 1b). No RNA could be detected in the GFP^{3HBH} sample, irrespective of UV-crosslinking. We controlled our loading by immunoblotting a fraction of each of the sample, which we show underneath Bioanalyzer traces.

As we expected and as the reviewer also suspected, single FLAG purifications showed no signs of specific RNA enrichment (Supplementary Fig. 1b) and thus should be avoided or used in conjunction with PAGE and nitrocellulose-transfer as frequently done in PAR-CLIP and iCLIP experiments.

Related to point 1) c) : The very high correlation of peaks between biological replicates revealed with the use of the stringent peak calling method PEAKachu is nicely convincing. In addition, giving the correlation values between unrelated replicates – as done with JAMM peakfinder in figure 2c- would even strengthen this analysis.

We have now added the correlations of unrelated replicates on PEAKachu peaks. The average Spearman correlation between unrelated replicates is 0.19 compared to an average Spearman correlation of 0.92 for biological replicates.

Related to point 7) : Some figures still appear in a different order in the text versus the figure panels (see figure 2)

We reorganized some of the panels so that figures appear as they are referenced in the text.